# Systematic Preserflo MicroShunt Intraluminal Stenting for Hypotony Prevention in Highly Myopic Patients: A Comparative Study

**DOI:** 10.3390/jcm12041677

**Published:** 2023-02-20

**Authors:** Enrico Lupardi, Gian Luca Laffi, Antonio Moramarco, Piero Barboni, Luigi Fontana

**Affiliations:** 1Ophthalmology Unit, IRCCS Azienda Ospedaliero-Universitaria di Bologna, 40138 Bologna, Italy; 2Studio Oculistico d’Azeglio, 40123 Bologna, Italy

**Keywords:** glaucoma surgery, minimally invasive glaucoma surgery, intraocular pressure, Preserflo MicroShunt, suture stent, primary open-angle glaucoma, myopia, hypotony

## Abstract

Transient hypotony is the most common early complication after Preserflo MicroShunt (PMS) implantation. High myopia is a risk factor for the development of postoperative hypotony-related complications; therefore, it is advisable that PMS implantation in patients should be performed while employing hypotony preventive measures. The aim of this study is to compare the frequency of postoperative hypotony and hypotony-related complications in high-risk myopic patients after PMS implantation with and without intraluminal 10.0 nylon suture stenting. This is a retrospective, case–control, comparative study of 42 eyes with primary open-angle glaucoma (POAG) and severe myopia that underwent PMS implantation. A total of 21 eyes underwent a non-stented PMS implantation (nsPMS), while in the remaining eyes (21 eyes), PMS was implanted with an intraluminal suture (isPMS group). Hypotony occurred in six (28.57%) eyes in the nsPMS group and none in the isPMS group. Choroidal detachment occurred in three eyes in the nsPMS group; two of them were associated with the shallow anterior chamber and one was associated with macular folds. At 6 months after surgery, the mean IOP was 12.1 ± 3.16 mmHg and 13.43 ± 5.22 mmHg (*p* = 0.41) in the nsPMS and isPMS group, respectively. PMS intraluminal stenting is an effective measure to prevent early postoperative hypotony in POAG highly myopic patients.

## 1. Introduction

Preserflo MicroShunt (PMS) (Santen Inc., Miami, FL, USA) implantation with mitomycin C (MMC) is considered a safe and effective surgical procedure for glaucoma patients [1,2,3,4,5]. However, postoperative complications may develop, leading to prolonged recovery and surgical failure in some cases. Transitory hypotony is the most common postoperative complication occurring in 11.1% to 39% of the eyes [1,2,3,4,6,7], complicated by choroidal detachment in 8.7% to 10.6% and the shallow anterior chamber in 2% to 13% of the eyes [3,4,6,7]. Despite the relatively high frequency of postoperative hypotony after PMS implantation, preventive measures have not been proposed yet. In this regard, the systematic intraluminal stenting and tube ligature for the Baerveldt and Molteno valve implantation [8,9,10,11] and releasable sutures for trabeculectomy [12,13] have proved effective in decreasing the risk of early postoperative hypotony. After filtration surgery, hypotony is frequently transitory and compatible with a good visual outcome, but in highly myopic patients (>6D), hypotony is often complicated with choroidal detachment, macular folds, suprachoroidal hemorrhage, tube-cornea touch, and corneal endothelial decompensation, which may result in delayed visual recovery, reintervention such as bleb revision, vitrectomy, and keratoplasty, and permanent loss of vision in some cases [14,15,16,17,18,19,20,21,22]. For these reasons, hypotony preventive measures in PMS implantation might help to reduce the risk of harmful postoperative complications. This study aims to (a) describe a technique of PMS intraluminal stenting using a 10.0 nylon suture, (b) assess its efficacy in reducing the risk of early postoperative hypotony and its complications in myopic patients, and (c) evaluate its effect on intraocular pressure (IOP) control within the first six months.

## 2. Materials and Methods

### 2.1. Study Design

We conducted a retrospective comparative cohort study on consecutive patients who underwent PMS implantation with or without intraluminal stenting at the Ophthalmology Unit of the IRCCS Policlinico S. Orsola di Bologna (Bologna, Italy), between January 2019 and April 2022. The collection of clinical data was conducted according to the principles of the Declaration of Helsinki and was given local regulatory approval by the AVEC regional committee. Written informed consent was obtained from all patients.

### 2.2. Participants

The study included adult patients who were diagnosed with primary open-angle glaucoma (POAG) or pseudoexfoliative glaucoma and severe myopia with inadequately controlled IOP, despite maximally tolerated medical therapy, or who were intolerant to glaucoma medications, requiring surgical intervention. Patients were divided into two groups according to the surgical technique adopted: non-stented PMS implantation (nsPMS) and PMS implantation with intraluminal stenting (isPMS). Procedures with anterior chamber bleeding and/or complicated device implantation were excluded from the study.

Severe myopia was defined as a spherical equivalent equal to or greater than −6D at manifest refraction with an axial length equal to or greater than 25 mm, measured with IOL MASTER 700 (Zeiss, Oberkochen, Germany). Patients with normal-tension glaucoma, neovascular glaucoma, pigmentary glaucoma, uveitic glaucoma, steroid-induced glaucoma, glaucoma associated with ocular dysgenesis, congenital glaucoma, history of ocular trauma in the study eye, and scleral buckling were excluded from this study.

### 2.3. Preoperative Assessment

Before surgery, a complete ophthalmologic evaluation was conducted, including slit-lamp examination, gonioscopy to confirm open-angle with visible trabecular meshwork, IOP measurement by Goldman applanation tonometer (Haag-Streit, Koeniz, Switzerland), and visual field examination using the Humphrey Visual Field 30-2 (HFA II 750, Zeiss-Humphrey Systems, Dublin, CA, USA). Baseline demographic information such as sex and age were documented, together with the number of glaucoma medications, previous ocular surgeries, and complications.

### 2.4. Surgical Technique

Surgical procedures were performed by two experienced glaucoma surgeons (G.L. and L.F.). PMS implantation was carried out following the previously published recommendation [3]. Briefly, after performing a limbal conjunctival and Tenon’s peritomy in the superonasal quadrant, MMC 0.03% was applied with three soaked surgical sponges on the bare sclera and under the conjunctival flap for three minutes and then thoroughly rinsed. Starting at 3 mm from the limbus, a 2 mm length scleral tunnel was created using a 1 mm micro knife, then a 25 gauge needle was used to enter into the anterior chamber just above the scleral spur. The PMS was then inserted into the scleral tunnel parallel to the iris plane incarcerating the device wings into the tunnel. PMS function was confirmed by the observation of aqueous outflow from the external end of the device. Upon confirmation of the device functioning, a 10.0 nylon suture was threaded into the PMS lumen through the external orifice (Figure 1A) and advanced until it reached the internal end (Figure 1B). The free end of the suture was then buried into a limbal corneal groove in order to provide easy access for its removal (Figure 1C). In the end, the conjunctiva was sutured at the limbus with a 7.0 vicryl suture (Figure 1D). Finally, a subconjunctival injection of 0.8 mL of betamethasone, 2 mg, and cefuroxime, 75 mg, was administered.

Postoperatively all patients were instructed to stop their previous glaucoma medication. Initial postoperative therapy included a topical association with netilmicin, 3 mg/mL, and dexamethasone, 1 mg/mL, six times per day and atropine, 10 mg/mL, eye drops BID for one week. After the first two weeks, therapy was reduced to only dexamethasone, 1.5 mg/mL, eye drops QID for 1 month and then progressively tapered during the first three months.

### 2.5. Intraluminal Stent Removal

Stent removal was carried out at the slit lamp under topical anesthesia. For this procedure, the corneal end of the intraluminal 10.0 nylon suture was grasped and pulled out from the PMS lumen. The arbitrary criteria for stent removal were IOP > 18 mmHg within the first two follow-up visits and IOP > 12 mmHg starting from the third postoperative visit to the 45th day. If the stent was still in place on the 45th day after surgery, it was trimmed at the emergence from the conjunctiva and the free segment was removed in order to avoid further exposure of the suture. These criteria were developed without literature references, following the clinical experience of the surgeons. The aim was to balance the need to remove the stent as early as possible, to avoid minimizing the hypotensive potential of the device, and the risk of postoperative hypotony.

### 2.6. Postoperative Assessment

Postoperative data were collected on day 1, week 1, 2, and 3, and month 1, 2, 3, and 6. Slit lamp examination, IOP measurements, number of glaucoma medications including oral acetazolamide, and complications were recorded at each postoperative visit. Postoperative IOP was managed according to the surgeon’s preferences and experience. IOP one week after stent removal was also collected.

### 2.7. Outcomes

The primary outcome was the frequency of hypotony, defined as an IOP < 6 mmHg at two consecutive study visits.

The secondary outcomes were as follows:-Number of complications.-IOP control at each study visit. Overall success was defined as an IOP decrease ≥30% from baseline and an absolute IOP ≤ 18 mmHg, which was considered complete without the use of glaucoma medications or as qualified with the use of antiglaucoma drugs after stent removal. Additional glaucoma surgery requiring a return to the operating room for persistent high IOP despite medical therapy was considered as failure. Intraluminal stent removal was not considered a glaucoma reoperation.-Number of stent removals and IOP one week following stent removal.-Number of antiglaucoma medications after surgery.

### 2.8. Statistical Analysis

The Shapiro–Wilk test was used to prove the normality of the data. Clinical and demographic data were expressed in terms of absolute number and percentage for categorical variables, and normal and non-normal continuous variables were reported as mean ± standard deviation (SD) and median with interquartile range, respectively. To compare means and proportions among groups, we used independent samples t-tests for normally distributed samples, the Mann–Whitney test for non-normal distribution, and χ2 (or Fisher’s exact test, where appropriate) tests for categorical variables. A value of *p* < 0.05 was considered significant for all tests. The confidence interval (CI) was 2-tailed and calculated considering a 0.95 confidence level. Snellen VA measurements were converted to logMAR equivalents for the purpose of data analysis. Statistical analysis was performed using SPSS for Windows (software version 25, IBM Corp., Armonk, NY, USA).

## 3. Results

Forty-two consecutive eyes (36 patients) with glaucoma and myopia underwent PMS implantation. Of these, 21 eyes out of the 16 patients underwent nsPMS, while 21 eyes out of 20 patients underwent isPMS implantation. All the patients completed a six month follow-up and were, therefore, included in the analysis. The baseline demographic and clinical characteristics are summarized in (Table 1). Sex was equally distributed, and age was comparable among the two groups. The mean spherical equivalent was −8.96 ± 5.5D in the nsPMS group and −10.81 ± 5D (*p* = 0.42) in the isPMS group, while the mean AXL was 28.34 ± 4.23 mm and 27.51 ± 3.97 mm (*p* = 0.81), respectively. Preoperative IOP was 26.47 ± 6.04 mmHg and 25.19 ± 6.49 mmHg (*p* = 0.5), respectively; the mean MD and PSD were comparable between the two groups.

Regarding previous operations, 9 and 15 patients in the nsPMS and isPMS groups, respectively, underwent surgery on the same eye before PMS implantation. In the nsPMS group, four patients were pseudophakic, two had cataract surgery combined with trabeculectomy, three had a deep sclerectomy, and one had a Xen45 gel stent (Allergan, an AbbVie company, North Chicago, IL, USA) implantation. In the isPMS group, eight had cataract surgery, three underwent combined cataract surgery and vitrectomy, two underwent combined cataract surgery and trabeculectomy, and one patient had cataract surgery and trabeculectomy without IOL implantation.

### 3.1. Primary Outcome

In the nsPMS group, transitory hypotony occurred in six (28.6%) eyes, lasting for two weeks in five eyes, and three weeks in one eye. Among those eyes, one had a previous deep sclerotomy, two were virgin eyes, and three of them were pseudophakic. In the isPMS group, none of the patients developed hypotony (Table 2).

### 3.2. Secondary Outcomes

#### 3.2.1. Complications

The postoperative complications differed significantly between the two groups with seven occurrences in the nsPMS group and one in the nsPMS group (*p* = 0.045) (Table 2). Hypotony-related complications, on the other hand, occurred only in the nsPMS group and choroidal detachment occurred in three cases: in two of them it was associated with a shallow anterior chamber, and in the other eye with macular folds. All cases were treated with atropine 1% eye drops and resolved within 15 days from the onset. In each group, a 3 mm hyphema developed in one eye from postoperative day one. Both cases were managed conservatively until their complete resolution, which occurred at week two.

#### 3.2.2. IOP Control

IOP values measured at each follow-up visit are shown in (Table 3). Lower mean IOP values were achieved in the nsPMS group compared to the isPMS group, reaching statistical significance at all the follow-up intervals during the first three weeks (Figure 2). At the 6-month visit, the mean IOP was 12.1 ± 3.16 in the nsPMS group and 11.86 ± 2.50 mmHg (*p* = 0.78) in the isPMS group, corresponding to a 46.70% and 49.01% reduction from baseline, respectively. Seventeen eyes (80.95%) in the nsPMS and sixteen eyes (76.19%) in the isPMS group achieved complete success (*p* > 0.9), while three eyes (14.29%) in the nsPMS and five eyes (23.81%) in the isPMS had an IOP > 18 mmHg at two consecutive visits following stent removal, requiring anti-glaucoma medications (*p* = 0.69), and were therefore classified as a qualified success. One patient (4.76%) in the nsPMS group was considered a failure due to a persistent high IOP and bleb fibrosis requiring a surgical reintervention at 6 months after the PMS implantation (Table 2).

#### 3.2.3. Intraluminal Stent

In 17 eyes (80.95%) the intraluminal stent was removed during the first month; the mean time of removal was 15.88 ± 5.85 days after surgery, and the mean IOP reduction after one week was 5.58 ± 1.09 mmHg. Pre-stent removal IOP was 15.94 ± 3.85 mmHg and after the removal was 10.35 ± 2.76 mmHg (*p* < 0.001). Three stents (14.29%) were removed at week one, obtaining an IOP reduction of 8, 15, and 11 mmHg. A total of nine stents (42.86%) were removed at 2 weeks with a mean IOP reduction of 4.36 ± 3.59 mmHg. Four stents (19.04%) were removed at 3 weeks, and one (4.8%) was removed one month after surgery with a mean IOP reduction of 5 ± 5.43 mmHg (Figure 3). In four (19.05%) eyes, the stent was removed because the IOP was higher than 18 mmHg, three of them (14.29%) at the first-week follow-up, and the other two one week later. Four eyes (19.05%) did not meet the criteria for stent removal at the end of follow-up and therefore, were left in place and trimmed on the 45th day, as described above.

#### 3.2.4. Number of Glaucoma Medications

Within the first six months after surgery, the mean number of glaucoma medications was 0.19 ± 0.51 in the nsPMS group and 0.33 ± 0.66 in the isPMS group. The mean reduction in the topical medication burden was 3.33 ± 0.8 and 3 ± 1.3 (*p* = 0.4), respectively (Table 2). Oral acetazolamide was not prescribed to any patients during the postoperative period.

## 4. Discussion

In the current study, we retrospectively examined the hypotony rate of consecutive highly myopic patients undergoing PMS implantation with and without intraluminal stenting. The systematic preventive approach resulted in the absence of postoperative hypotony cases in the isPMS group compared to the 28.6% presence in the nsPMS group, which is in accordance with the previously reported data in the literature [1].

Although the effect of the 10.0 nylon stent for outflow restriction as a treatment for postoperative prolonged hypotony was previously described [23], this is the first study to investigate the efficacy of PMS flow restriction in high-risk myopic eyes and its preventive effect on the early hypotony rate. In vitro studies proved that the PMS does not offer hydraulic resistance to outflow similarly to other non-valved shunts, for example, Molteno and Baerveldt [24,25,26]; therefore, the only opposition to excessive outflow is the filtration bleb pressure, which, in the early postoperative period, is minimal due to the absence of conjunctival and Tenon fibrosis [25]. It follows that a major problem of non-valved, bleb-forming, shunt implantation is the early outflow control [8,9,10,11,24,27,28], and early hypotony is a relatively frequent complication compared to late hypotony, which occurs more rarely. Following filtration surgery, hypotony may occur more frequently in highly myopic eyes due to their anatomic characteristics, which include a thinner and less rigid scleral wall. Furthermore, in these eyes, there is an increased risk of sight-threatening hypotony-related complications such as choroidal detachment, macular folds, and suprachoroidal hemorrhage [14,15,16,17,18,19,20,21,22]. For these reasons, preventing hypotony in these eyes should be considered a priority. In our study, hypotony was complicated by choroidal detachment in half of the cases, and other related complications such as shallow anterior chamber and macular folds were recorded in the nsPMS group, while the absence of hypotony in the isPMS group prevented their occurrence in these eyes.

With the minimal 10.0 nylon suture caliber (20 µm) and PMS lumen (70 µm), stent insertion does not require specific surgical skills, it usually does not represent a challenge for the surgeon, and it extends the intraoperative time by only a few minutes. In our experience, baring the external end of the suture in a corneal groove facilitates the postoperative management of the stent, avoiding the need for invasive procedures for its removal [8], while minimizing the risk of unintentional suture pullout and patient discomfort.

In this study, the effect on the outflow restriction of the 10.0 nylon is testified by the higher mean IOP of the isPMS group in the first three weeks (TAB3.) and the significant difference between pre and post-removal IOP. On the other hand, the decreasing trend in the IOP reduction indicates the effect of bleb fibrosis on the IOP drop after stent removal (*p* = 0.043, R^2^ = 0.25) (Figure 3). For these reasons, the timing of the stent removal is also of critical importance. Following our criteria, we managed to prevent post-removal hypotony, which might otherwise happen if the stents were removed earlier or with a lower IOP.

Regarding IOP control, complete success was reached by most of the eyes in both groups without a statistical difference (*p* > 0.99). Furthermore, a comparable reduction in anti-glaucoma eye drops use was achieved in both groups (*p* = 0.33). These results may suggest that, in contrast with the delayed IOP lowering effect experienced in non-valved tubes with flow restriction [8,29,30], the 10.0 intraluminal stent effectively increases IOP without affecting the efficacy of PMS filtration at 6 months.

All surgeries were carried out with 3 min of MMC 0.03% episcleral application before performing the scleral tunnel. The optimal MMC concentration in patients at risk for hypotony-related complications is still debated; some studies have shown an increased risk with a higher MMC concentration after trabeculectomy [31,32]. However, recent recommendations were not conclusive regarding the MMC concentration to use in PMS surgery and left the final decision to the surgeon’s own experience [5]. In our opinion, in PMS implantation, an intraluminal stent insertion might be preferable to a lower MMC concentration to avoid early hypotony. Firstly, because suture stent is reversible, and secondly, because a higher MMC concentration might prevent long-term surgical failure. Nevertheless, further studies are needed to confirm this hypothesis.

There are several inherent limitations of this study, including its retrospective design, lack of power calculation, and limited follow-up, which are all inherent to the process of acquisition of any new surgical technique. The small sample of eyes does not permit a sub-analysis of hypotony-related complications or the effect of phacoemulsification prior to PMS implantation. The inclusion of fellow eyes may generate coupling errors in statistical analysis; the short follow-up does not enable long-term efficacy predictions; and the inclusion of only severely myopic eyes prevents the application of these results to standard eyes. Furthermore, only the effect of the 10.0 nylon suture was investigated; more comparative studies are needed to assess the effect of different stent sizes and materials on the hypotony rate.

## 5. Conclusions

Our study suggests that a 10.0 nylon stent is effective in limiting the PMS flow after surgery, lowering the risk of hypotony in highly myopic eyes. This technique shows a good safety profile and does not affect IOP lowering six months after surgery. Our results warrant the systematic use of intraluminal 10.0 nylon stents during PMS implantation in severely myopic eyes to prevent hypotony-related complications.

## Figures and Tables

**Figure 1 jcm-12-01677-f001:**
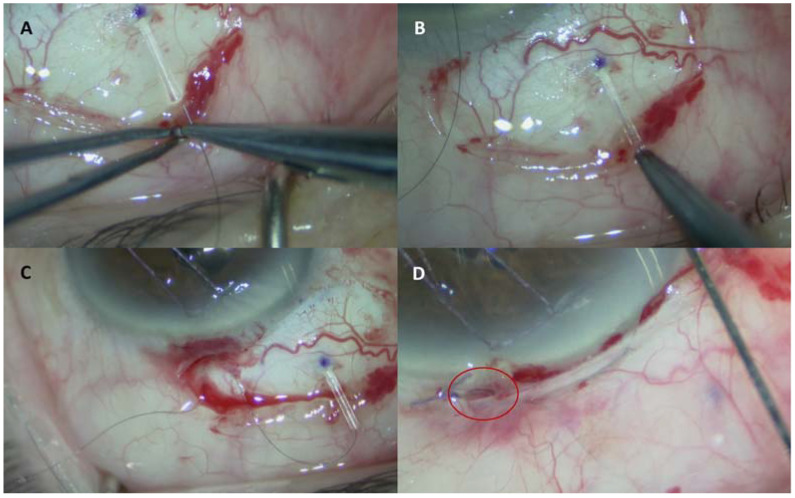
Intraoperative images showing the surgical steps of intraluminal stent insertion. (**A**) A 10.0 nylon suture is inserted in the outer lumen of the PMS. (**B**) The suture is advanced inside the PMS lumen to reach the inner end. (**C**) The external part of the suture is buried in a limbal corneal grove. (**D**) Image showing the suture loop protruding from the conjunctiva enables easy postoperative access.

**Figure 2 jcm-12-01677-f002:**
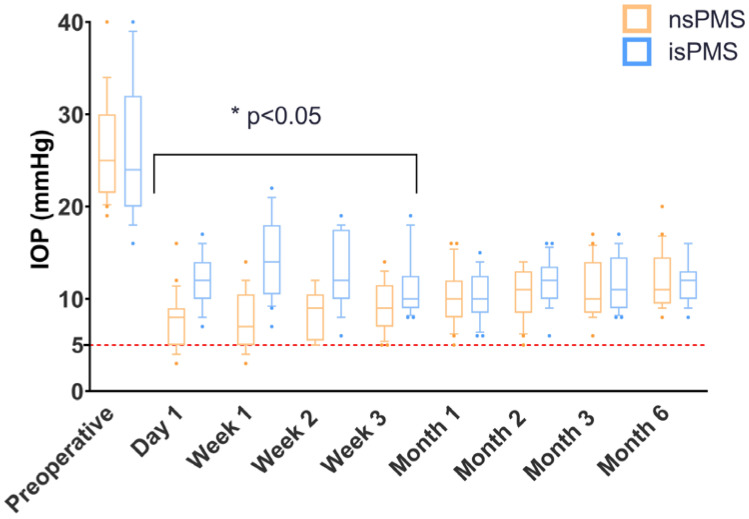
Box and whiskers plot showing the summary of postoperative intraocular pressure (IOP) measurements at each time point for each group. The height of the box represents the lower and upper interquartile range (50% of the range), and the ends of the whiskers extend to 80% of the range (10% to 90%). The line bisecting each box represents the median IOP. The horizontal red dashed line marks the lower IOP limit associated with hypotony (<6 mm Hg). The median IOP is also shown beneath the line. nsPMS: non-stented Preserflo MicroShunt implantation. isPMS: intraluminal stenting Preserflo MicroShunt implantation. IOP: intraocular pressure.

**Figure 3 jcm-12-01677-f003:**
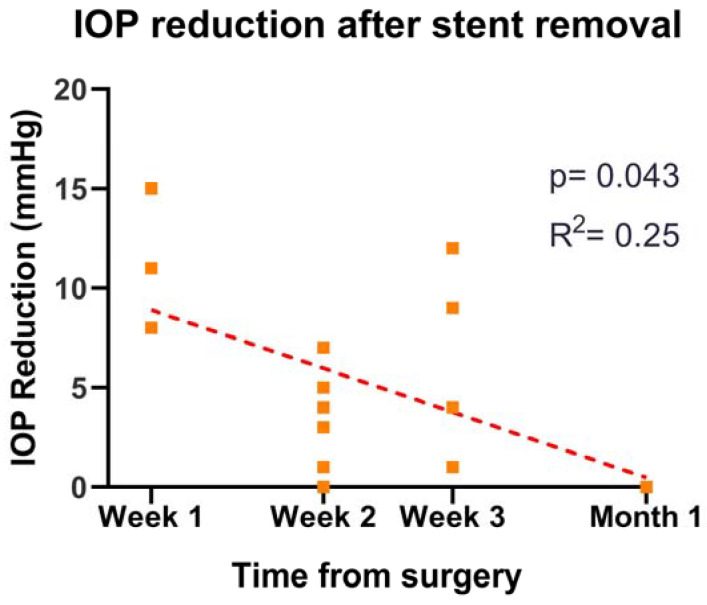
Summary of the relationship between IOP reduction one week after the stent removal and the time of removal. The squares show the IOP reduction in each eye one week after the stent removal. The red dashed line represents the linear regression between IOP reduction and time of removal. IOP: intraocular pressure.

**Table 1 jcm-12-01677-t001:** Baseline demographics and clinical characteristics of the study population.

	nsPMS ^1^ *n* = 21	isPMS ^2^ *n* = 21	*p*
**Female (%)**	15 (71.42)	14 (66.67)	>0.99
**Age (mean** **±** **SD)**	61.76 ± 14.92	62.95 ± 13.49	0.92
**Primary glaucoma diagnosis (%)**			
Open angle	16 (76.19)	14 (66.67)	0.167
Pseudoexfoliative	5 (23.81)	7 (33.33)	
**Central corneal thickness (****µ****m) (mean** **±** **SD)**	526.74 ± 30.92	543.39 ± 36.47	0.2
**Number of previous incisional surgery (%)**	5 (23.81)	3 (14.29)	0.69
**Number of previous phacoemulsification (%)**	6 (28.57)	14 (66.67)	0.03
**Preoperative measures (mean ± SD or median (IQ))**			
IOP (mmHg)	26.47 ± 6.04	25.19 ± 6.49	0.5
BCVA (LogMAR)	0.52 ± 0.52	0.29 ± 0.29	0.08
Spherical equivalent (D)	−8 [−6, −11]	−9 [−7, −13]	0.45
Axial length (mm)	28.34 ± 4.23	27.51 ± 3.97	0.09
MD	−15.77 ± 9.73	−15.50 ± 9.64	0.71
PSD	7.63 ± 4.37	9.25 ± 6.01	0.54
**IOP-lowering medications (median** **(IQ)****)**	3 [3, 4]	4 [2.5, 4]	0.85
**Oral acetazolamide (%)**	15 (71.43)	12 (57.14)	0.52

^1^ nsPMS: non-stented PMS implantation; ^2^ isPSM: intraluminal stent PMS implantation.

**Table 2 jcm-12-01677-t002:** Postoperative outcomes.

	nsPMS ^1^ *n* = 21	isPMS ^2^ *n* = 21	*p*
**Hypotony (%)**	6 (28.57)	0	
**IOP ^3^ control (%)**			
Complete	17 (80.95)	16 (76.19)	>0.99
Qualified	3 (14.29)	5 (23.81)	0.69
Failed	1 (4.76)	0	
**Number of glaucoma medication (mean ± SD)**			
Mean reduction at six months	3.24 ± 0.70	2.90 ± 1.37	0.33
Month 1	0.05 ± 0.22	0.10 ± 0.30	0.56
Month 2	0.14 ± 0.48	0.10 ± 0.30	0.70
Month 3	0.19 ± 0.51	0.19 ± 0.51	
Month 6	0.19 ± 0.51	0.33 ± 0.66	0.44
**Postoperative complications (%)**			
Total	7 (33.33)	1 (4.76)	0.045
Choroidal detachment	3 (14.29)	0	
Shallow anterior chamber	2 (9.52)	0	
Macular folds	1 (4.76)	0	
Hyphema	1 (4.76)	1 (4.76)	

^1^ nsPMS: non-stented PMS implantation; ^2^ isPSM: intraluminal stent PMS implantation; and ^3^ IOP: intraocular pressure.

**Table 3 jcm-12-01677-t003:** Postoperative IOP values and IOP reduction after ns PMS and is PMS implantation at each follow-up.

	Postoperative IOP ^3^ (Mean ± SD)			Postoperative IOP ^3^ Reduction from Baseline (Mean ± SD ^4^)		
	nsPMS ^1^	isPMS ^2^	*p*	nsPMS ^1^	isPMS ^2^	*p*
Day 1	7.24 ± 3.13	11.66 ± 2.81	<0.0001	19.00 ± 7.15	13.86 ± 7.47	0.039
Week 1	7.62 ± 3.15	14.52 ± 4.35	<0.0001	13.86 ± 6.84	10.67 ± 9.06	0.003
Week 2	8.19 ± 2.64	12.81 ± 4.04	0.0002	10.67 ± 6.9	12.38 ± 8.2	0.027
Week 3	9.38 ± 2.52	11.52 ± 3.36	0.043	12.38 ± 6.50	13.67 ± 7.05	0.135
Month 1	10.19 ± 2.91	10.43 ± 2.69	0.96	13.67 ± 6.22	14.76 ± 6.89	0.54
Month 2	10.62 ± 2.82	11.67 ± 2.48	0.32	14.76 ± 5.86	13.52 ± 6.23	0.27
Month 3	11 ± 3.07	11.76 ± 2.94	0.41	13.52 ± 6.37	13.43 ± 6.27	0.36
Month 6	12.1 ± 3.16	11.86 ± 2.5	0.78	13.43 ± 5.22	13.33 ± 6	0.55

^1^ nsPMS: non-stented PMS implantation; ^2^ isPSM: intraluminal stent PMS implantation; ^3^ IOP: intraocular pressure; and ^4^ SD: standard deviation.

## Data Availability

All data are available from the corresponding author upon request.

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
