# Peer review of "Systematic Preserflo MicroShunt Intraluminal Stenting for Hypotony Prevention in Highly Myopic Patients: A Comparative Study"

_jcm, 2023, doi:10.3390/jcm12041677_

Round 1
Reviewer 1 Report
“Systematic Preserflo-Microshunt Intraluminal Stenting for Hypotony Prevention in Highly Myopic Patients: A Comparative Study” introduced a hypotony preventive measures after Preserflo-Microshunt implantation. Authors showed PMS implantation with intraluminal 10.0 nylon suture stenting is effective to prevent transient hypotony after surgery.
Minor
1. Line 88 to 89; Upon confirmation of the device functioning, when is PMS surgery was planned, a 10.0 nylon suture was threaded into the PMS lumen through the external orifice (FIG1. A)
Authors can delete “when isPMS surgery was planned,”
2. Line 111 to 112; The arbitrary criteria for stent removal were IOP > 18mmHg or IOP > 12mmHg at any postoperative control visit starting from two weeks after surgery.
This sentence is very confusing. Did authors remove the nylon stent, when the IOP is 15 mmHg?
3. Authors mentioned that HRCs means Hypotony-related complications in line 183. “HRCs” is unfamiliar abbreviation. HRCs appears a little in the discussion section. Please use “hypotony-related complications” instead of HRCs.
Line 284 to 301; authors use “HRC” not “HRCs”.
Author Response
- Line 88 to 89; Upon confirmation of the device functioning, when is PMS surgery was planned, a 10.0 nylon suture was threaded into the PMS lumen through the external orifice (FIG1. A)
Authors can delete “when isPMS surgery was planned,”
- Response: Thank you for the comment, we have deleted the sentence from the manuscript.
- Line 111 to 112; The arbitrary criteria for stent removal were IOP > 18mmHg or IOP > 12mmHg at any postoperative control visit starting from two weeks after surgery.
This sentence is very confusing. Did authors remove the nylon stent, when the IOP is 15 mmHg?
- Response : During the first two follow-up visits, the stent was removed whenever IOP was higher than 18mmHg. Starting from the third follow-up visit (the second week after surgery), the stent was removed if IOP was higher than 12mmHg.
We agree with the reviewer that the referred sentence might lead to confusion and misunderstandings, therefore we changed it with: “. The arbitrary criteria for stent removal were IOP > 18mmHg within the first two follow-up visits and IOP > 12mmHg starting from the third postoperative visit to the 45th day”
- Authors mentioned that HRCs means Hypotony-related complications in line 183. “HRCs” is unfamiliar abbreviation. HRCs appears a little in the discussion section. Please use “hypotony-related complications” instead of HRCs.
Line 284 to 301; authors use “HRC” not “HRCs”.
- Response: We corrected the mistake, and substitute each acronym with the extended form.
Reviewer 2 Report
1. This is quite a practical paper which will be referenced by future clinicians.
2. Page 1 line 13-14, in the abstract, the abbreviation “HRCs” should be added after “hypotony related complications”.
3. Please provide manufacturer/location details for the Preserflo-Microshunt in the method section.
4. “The arbitrary criteria for stent removal were IOP>18mmHg or IOP>12mmHg at any postoperative control visit starting from two weeks after surgery. If the stent was still in place on the 45th day after surgery, it was trimmed.” Were there any reference support the IOP (18mmHg/12mmHg) and time point (2 weeks/45days) determination in this criteria? If this was from personal experience, can the authors comment this further?
5. “21 eyes of 16 patients underwent nsPMS while 21 eyes of 20 patients underwent isPMS implantation.” As prognosis following surgical intervention between fellow eyes tend to be similar, this may cause coupling effect in the statistical analysis. Please clarify this statistical limitation in the manuscript.
6. One of the limitations is the difference in number of previous phacoemulsification. The isPMS group has higher previous phacoemulsification numbers. Phacoemulsification has been proved to have influence on POAG IOP level especially in early post-operation stage.
7. As stated in the limitation, follow up is relatively short at 6 months. Why choose 6 months for the maximum follow up? It would be more meaningful if long term IOP changes and complications be reported.
Author Response
- This is quite a practical paper which will be referenced by future clinicians.
- Response: We thank the reviewer for the kind comment and interest in our study
- Page 1 line 13-14, in the abstract, the abbreviation “HRCs” should be added after “hypotony related complications”.
- Response: As suggested by Reviewer 1, we substituted each HRC acronym with the extended form.
- Please provide manufacturer/location details for the Preserflo-Microshunt in the method section.
- Response: We thank the reviewer for highlighting missing information, we have added the required details in the manuscript (lines 33)
- “The arbitrary criteria for stent removal were IOP>18mmHg or IOP>12mmHg at any postoperative control visit starting from two weeks after surgery. If the stent was still in place on the 45th day after surgery, it was trimmed.” Were there any reference support the IOP (18mmHg/12mmHg) and time point (2 weeks/45days) determination in this criteria? If this was from personal experience, can the authors comment this further?
-Response: We thank the reviewer for the comment. The criteria were developed after a brief initial practical experience on PMS intraluminal stenting in normal eyes. We realized that the healing and fibrosing process of the PMS subtenonian bleb is faster than that of traditional valves with a more posterior filtration, in which stent removal is usually carried out later. It is probably due to the less invasive dissection of the conjunctiva and the higher vascularization of the tissues. As stated in the manuscript, these were arbitrary criteria, therefore developed without literature references, balancing the need to remove the stent as early as possible with the need to minimize the risk of postoperative hypotony.
We added the sentence “These criteria were developed without literature references, following the clinical experience of the surgeons. The aim was to balance the need to remove the stent as early as possible, to avoid minimizing the hypotensive potential of the device, and the risk of postoperative hypotony.”
- “21 eyes of 16 patients underwent nsPMS while 21 eyes of 20 patients underwent isPMS implantation.” As prognosis following surgical intervention between fellow eyes tend to be similar, this may cause coupling effect in the statistical analysis. Please clarify this statistical limitation in the manuscript.
- Response: We agree with the reviewer regarding the coupling error in the statistical analysis and the interpretation of the results. we have added this as a limitation to the study. (lines 318-319)
- One of the limitations is the difference in number of previous phacoemulsification. The isPMS group has higher previous phacoemulsification numbers. Phacoemulsification has been proved to have influence on POAG IOP level especially in early post-operation stage.
-Response: We agree with the reviewer that previous phacoemulsification might have an effect on early postoperative IOP control. However, to date, no studies have investigated the effect of previous cataract surgery on PMS implantation. Moreover, Fili et. al (J Curr Ophthalmol 2022, 10.4103/joco.joco_298_21), reported no differences in postoperative IOP control and complications between phaco +PMS implantation and PMS standalone implantation in phakic eyes. Unfortunately, our sample is not powered enough to provide a sub-analysis on phakic vs pseudophakic eyes. We added the following sentence to the limitation of the study (lines 317-319)
“The small sample of eyes does not permit a sub-analysis of hypotony-related complications, and the effect of phacoemulsification prior to PMS implantation.”
- As stated in the limitation, follow up is relatively short at 6 months. Why choose 6 months for the maximum follow up? It would be more meaningful if long term IOP changes and complications be reported.
- Response: We thank the reviewer for the comment. The aim of the study was to compare frequency of postoperative hypotony between the two groups. We considered that postoperative hypotony usually occurs within the first 3 months after surgery and more rarely afterwards. For this reason we choose the 6 months interval as reasonable time period to measure our principle outcome. Regarding secondary outcomes, we agree with the reviewer that a longer follow-up might be more meaningful and some of these outcomes will be addressed again in future studies.
Reviewer 3 Report
The authors did an interesting, detailed comparison in terms of risk and frequency for hypotony in patients with high myopia after implementing modification for surgical treatment by Preserflo-Microshunt (PMS) implantation where some of the patients had stent while the others didn’t.
The author did a detailed statistical analysis that showed good difference that is helpful for glaucoma surgeons, however, minor changes are needed.
11. Abstract, please define HRCs.
22. Methods, authors need to define uncomplicated PMS implantation surgery.
33. Methods: The authors included patients with primary open-angle glaucoma, but later on in the results, they showed that in Table 1, more than 25% of participants had Psudoexpholiation which is a cause for secondary open-angle glaucoma. The authors need either to modify the inclusion criteria in the methods or to exclude the eyes with pseudoexfoliation.
44. Results: Can the authors say report about the possible risk of infection or inflammation after the surgery? Was there any difference between the 2 groups? Theoretically, we expect that the stent may increase the risk of infection or ocular inflammation.
55. Results: The authors mentioned the terms complete success and qualified success. These 2 terms need to be defined in the methods.
66. Discussion: The authors mentioned the limitations of their study. I hope they can report longer terms outcomes in the future.
Author Response
- Abstract, please define HRCs.
- Response: As suggested by Reviewer 1, we substituted each HRC acronym with the complete form.
- Methods, authors need to define uncomplicated PMS implantation surgery.
- Response: Thank you for highlighting a missing part in the manuscript, we added the description of the uncomplicated PMS implantation surgery in the methods, in lines 67-69
“Procedures with anterior chamber bleeding and/or complicated device implantation were excluded from the study”
- Methods: The authors included patients with primary open-angle glaucoma, but later on in the results, they showed that in Table 1, more than 25% of participants had Pseudoexfoliation which is a cause for secondary open-angle glaucoma. The authors need either to modify the inclusion criteria in the methods or to exclude the eyes with pseudoexfoliation.
- Response: We thank the reviewer for the correction, we have modified the inclusion criteria in order to include pseudoexfoliative glaucoma in the study (line 65)
- Results: Can the authors say report about the possible risk of infection or inflammation after the surgery? Was there any difference between the 2 groups? Theoretically, we expect that the stent may increase the risk of infection or ocular inflammation.
- Response: We agree with the reviewer that a higher risk of bleb infection and intraocular inflammation might result from the partially exposed 10.0 nylon. To decrease the risk of contamination we trimmed the exposed end of the stent on the 45th day after surgery, as mentioned in the manuscript (lines 123-125). To date, we have never witnessed such complications; therefore, we cannot report differences between the two groups. In our opinion, since those are rare complications of PMS implantation, our study may not be powered to highlight differences between groups. Future larger studies may address the risk of postoperative infection and inflammation after stent insertion.
- Results: The authors mentioned the terms complete success and qualified success. These 2 terms need to be defined in the methods.
- Response: The definition of overall success, complete success, and qualified success as well as failure are mentioned in the “outcomes” section of the methods as the second point of the “secondary outcomes”. (lines 140-145)
- Discussion: The authors mentioned the limitations of their study. I hope they can report longer terms outcomes in the future.
- Response: Thank you for the comment, and your interest in the study. We are currently following our patients and collecting more data in order to publish the results at one year of follow-up.